# Comparing Methods to Select Candidates for Re-Genotyping to Impute Higher-Density Genotype Data in a Japanese Black Cattle Population: A Case Study

**DOI:** 10.3390/ani13040638

**Published:** 2023-02-11

**Authors:** Shinichiro Ogawa, Atsushi Zoda, Rino Kagawa, Rui Obinata

**Affiliations:** 1Division of Meat Animal and Poultry Research, Institute of Livestock and Grassland Science, National Agriculture and Food Research Organization (NARO), Tsukuba 305-0901, Japan; 2Research and Development Group, Zen-Noh Embryo Transfer Center, Kamishihoro 080-1407, Japan

**Keywords:** high-density genotyping, imputation, Japanese Black cattle, pedigree, reference population, single-nucleotide polymorphism

## Abstract

**Simple Summary:**

The effectiveness of selecting individuals based on pedigree information (MCA) and genomic information (MCG) as a reference population for genotype imputation was assessed using a Japanese Black cow population. Pedigree depth to construct the pedigree-based additive genetic relationship matrix (**A** matrix) affected the results of MCA, and the method of calculating the genomic relationship matrix (**G** matrix) affected the results of MCG. For the studied population, MCG appeared to be a better choice than MCA to select cows as a reference population for higher-density genotype imputation to perform genomic prediction and a genome-wide association study.

**Abstract:**

As optimization methods to identify the best animals for dense genotyping to construct a reference population for genotype imputation, the MCA and MCG methods, which use the pedigree-based additive genetic relationship matrix (**A** matrix) and the genomic relationship matrix (**G** matrix), respectively, have been proposed. We assessed the performance of MCA and MCG methods using 575 Japanese Black cows. Pedigree data were provided to trace back up to five generations to construct the **A** matrix with changing the pedigree depth from 1 to 5 (five MCA methods). Genotype information on 36,426 single-nucleotide polymorphisms was used to calculate the **G** matrix based on VanRaden’s methods 1 and 2 (two MCG methods). The MCG always selected one cow per iteration, while MCA sometimes selected multiple cows. The number of commonly selected cows between the MCA and MCG methods was generally lower than that between different MCA methods or between different MCG methods. For the studied population, MCG appeared to be more reasonable than MCA in selecting cows as a reference population for higher-density genotype imputation to perform genomic prediction and a genome-wide association study.

## 1. Introduction

Japanese Black cattle are a breed famous for excellent meat quality (e.g., [1,2,3]). For this breed, it has been important to improve not only carcass characteristics including the degree of marbling but also traits relating to reproductive efficiency, feed efficiency, and meat quality [4,5,6]. Introducing a genomic prediction (GP) scheme is expected to contribute to further efficient genetic improvement of this breed, and GP for carcass traits (e.g., [7,8,9]) and other important traits (e.g., [10,11,12]) has been studied. Recently, we assessed the performance of pedigree-based breeding-value evaluation for superovulatory response traits in Japanese Black cattle [13,14,15]. As the superovulatory response traits are sex-limited traits and their heritabilities were estimated to be lower than those of carcass traits [14], it might be more desirable to introduce GP for the superovulatory response traits.

Under conditions where it is difficult and expensive to immediately collect genotype information on high-density single-nucleotide polymorphism (SNP) markers for all individuals, accurate genotype imputation from low-density genotypes to high-density genotypes is crucial for performing cost-effective GP and genome-wide association study (GWAS) (e.g., [16,17,18]). Studies on genotype imputation using commercial SNP markers have been performed in Japanese Black cattle. For example, Ogawa et al. [19] assessed the performance of imputing genotype data from low-density SNP genotypes (<50 K) in GP for carcass traits. In addition, Ogawa et al. [20] and Takeda et al. [21] compared the performance of GP with real genotype information on ~30,000 SNPs genotyped with the Illumina 50 K chip (Illumina Inc., San Diego, California USA) with imputed genotype information from low-density genotypes on ~550,000 SNPs, which can be genotyped with the Illumina HD chip, for carcass and feed efficiency traits, respectively. When the size of the reference population was approximately 400, the imputation from low-density genotype information using Beagle software (v3.3.2 or v4.0) [22] appeared to be completely accurate for cost-effective GP [23,24].

Most of the studies on GP in Japanese Black cattle have used tens of thousands of SNP markers (e.g., [19,20,21]), and we have been promoting genotyping using the Illumina 50 K chip in order to perform GP for superovulatory response traits. Meanwhile, to improve the performance of GP and GWAS by increasing the marker density via genotype imputation, a strategy to re-genotype some individuals with higher-density SNP chips and use them as a reference population appears to be available. Several studies have investigated strategies to choose individuals as a reference population for genotype imputation (e.g., [25,26,27]). Yu et al. [26] proposed two approaches, namely, selecting a candidate to minimize the conditional additive genetic variance of the target animals using either the pedigree-based additive genetic relationship matrix (**A** matrix) (MCA) or genomic relationship matrix (**G** matrix) (MCG). These approaches use **A** and **G** matrices as sources of information on the genetic structure of the population, so their performance may be population-dependent. On the other hand, the content of the **A** matrix was also shown to be affected by the quality of pedigree information, such as the number of generations traced back. Moreover, different methodologies were proposed to calculate the **G** matrix, including methods 1 and 2 proposed by VanRaden [28]. These might affect the results of MCA and MCG methods. In this study, aiming at future efficient re-genotyping to produce higher-density genotype information, we assessed the MCA and MCG methods while varying both the pedigree depth used to construct the **A** matrix and the method used to calculate the **G** matrix in our Japanese Black donor cows.

## 2. Materials and Methods

### 2.1. Ethics Statement

Animal Care and Use Committee approval was not needed because the data were obtained from existing databases.

### 2.2. Pedigree and SNP Genotype Data

A total of 575 Japanese Black cows, who were donors for superovulation treatments and born between 2000 and 2015, reared in the Zen-noh Embryo Transfer Center (Kamishihoro, Hokkaido) were used. Most of them were also part of 1546 Japanese Black donor cows with superovulation performance records of their own used in our previous studies [13,14,15]. The donor cows consisted of individuals introduced from the markets and their (grand)daughters. Genotype information of the 575 cows was obtained using the Illumina BovineSNP50 v2 BeadChip. Quality control (QC) was performed on 52,524 SNPs located on each of the 29 *Bos taurus* autosomes after updating the annotation to ARS-UCD1.2 using the LiftOver tool (https://genome.ucsc.edu/cgi-bin/hgLiftOver) (accessed on 19 October 2022) out of the 52,524 SNPs. For each SNP, the criteria of QC were set as minor allele frequency >0.01, call rate >0.95, and *p*-value > 0.001 for Hardy–Weinberg equilibrium (HWE) test. For each cow, the criterion of QC was set as call rate >0.95. After quality control, missing genotypes were filled using Beagle v3.3.2 [22]. Finally, genotype information on 36,426 SNPs on the Illumina BovineSNP50 v2 BeadChip was used. Pedigree data were constructed by tracing back up to five generations from the 1874 donor cows, including the 575 cows with genotype information. Finally, in this study the number of individuals included in the pedigree data was 4684 (Table 1).

### 2.3. Selection Based on MCA and MCG Methods

We iteratively selected individual(s) from the 575 cows by the MCA with pedigree-based relationship information and MCG methods with genome-based relationship information using the procedure described in [26] and exploited in previous studies [26,27]. Here, five matrices were used for MCA and two matrices were used for MCG.

The MCA method aims to minimize the trace of the following matrix **A**_11_* for each iteration [26,27]:$$A_{11}*=A_{11}-A_{12}A_{22}^{-1}A_{21}$$
where **A** is the block for the 575 cows of the entire **A** matrix; subscript 1 means a group of n cows already selected as candidates from the 575 cows; and subscript 2 means a group of the remaining 575—n cows. In MCA, the goal is to identify the population that minimizes the trace of the matrix **A**_11_*. When more than one cow was selected within a single iteration, all of the selected cows were simultaneously added to the candidate group. We provided the blocks for the 575 cows from **A** matrices varying in the number of generations tracing back. This was done to obtain information on the effect of the quality of pedigree information on the results of the MCA method.

For MCG, two **G** matrices for the 575 cows were used instead of the block of the **A** matrix. The first **G** matrix for the 575 cows, denoted as GV1, was calculated according to VanRaden’s method 1 as follows:$$\left(M-2p\right){\left(M-2p\right)}^{\prime }/\sum_{n=1}^{36,426}2p_{i}\left(1-p_{i}\right)$$
where **M** is the matrix containing the number of a counted SNP allele (0, 1, or 2); and **p** is the vector of the frequencies of counted SNP alleles. The second **G** matrix, GV2, was calculated according to VanRaden’s method 2 as follows:$$\left(M-2p\right)D{\left(M-2p\right)}^{\prime }/36426$$
where **D** is the diagonal matrix with its ith diagonal element as 2p_i_(1 − p_i_). Here, VanRaden’s methods 1 and 2 were selected because previous studies on GP in Japanese Black cattle also used **G** matrices calculated based on these methods (e.g., [19,20,21,29]).

### 2.4. Preparing Relationship Matrices for the 575 Cows from Pedigree and Genotype Information

We obtained the block **A** for the 575 cows based on the pedigree data containing 4684 individuals with the saveA22 option of BLUPF90+ software [30,31,32]. To assess the effect of pedigree depth on the results of the MCA method, the number of generations traced back from the 1874 cows for **A** matrix calculation, or setting for PED_DEPTH in parameter file, was changed from 1 to 5, and the blocks of resulting five **A** matrices for the 575 cows were denoted as A1, A2, …, and A5, respectively. For example, A1 was obtained as the block for the 575 cows from the **A** matrix constructed using pedigree information containing the 1874 cows and the 1521 parents (Table 1) when one generation was traced back. We also obtained two **G** matrices for the 575 cows, GV1 and GV2, with the saveG option of BLUPF90+. The whichG option was set to be 1 and 2 to obtain GV1 and GV2, respectively. Allele frequencies of the 36,426 SNPs were calculated using the 575 cows. For detailed explanation about calculating **A** and **G** matrices by BLUPF90+, see Lourenco et al. [33] for instance.

## 3. Results and Discussion

### 3.1. Comparing Matrices for MCA and MCG Methods

For the block of **A** matrix for the 575 cows, as expected, the average values of the diagonal and upper-triangular elements increased when the number of generations traced back was greater (Table 2), while the changes were negligible between A4 and A5. This was mainly due to an increased number of common ancestors when the number of generation traced back was greater. The pedigree-based inbreeding coefficient can be obtained as the diagonal element minus 1, and in this study, the pedigree-based inbreeding coefficients of the 575 cows ranged from 0 to 0.172 when the number of generations traced back was five. The base population of this study was likely more recent than those of previous studies. For example, Nomura et al. [34] calculated the inbreeding coefficients of Japanese Black bulls and heifers registered at the Wagyu Registry Association using pedigree data constructed by tracing back to ancestors born in 1945 or before. Honda et al. [35] calculated the inbreeding coefficients of a closed population of Japanese Black cattle in Hyogo prefecture using pedigree data traced back to the population in 1937 or before. Therefore, the pedigree-based inbreeding coefficients obtained here would be underestimated. To overcome this, a future task will be to develop a method to evaluate the degree of inbreeding for our Japanese Black donor cows using genome-wide DNA marker information.

For GV1 and GV2, the minimum value of diagonal elements was <1 and that of the upper-triangular elements was negative [28]. Note that genotypes of each SNP were weighted by the expected heterozygosity under HWE in VanRaden’s method 2 but not in method 1 [28], which produced the differences between GV1 and GV2. Previous studies have shown that the population structure of Japanese Black cattle could be captured with commercial SNP markers (e.g., [36,37,38]). Therefore, in this study, we concluded that both GV1 and GV2 calculated here could be used as the relationship matrix for MCG containing information on population structure of the 575 cows used. On the other hand, it might be also important that the **A** matrix is twice the matrix of identity-by-descent probabilities while the **G** matrix was the twice the matrix of identity-by-state probabilities [39]. VanRaden [28] stated that the genomic inbreeding coefficient, calculated as the value of diagonal element of the **G** matrix minus 1, is greater if the individual is homozygous for minor alleles than if it is homozygous for major alleles. Recently, various methods for inbreeding measures using genome-wide marker information have been proposed [40], and an investigation assessing the genomic inbreeding is also necessary for Japanese Black cattle.

### 3.2. Differences in the Number of Cows Selected in Each Iteration

In this study, all MCA methods sometimes selected more than one individual within an iteration; this was more prominent when using A1 and A2 for MCA (Figure 1). This is because some animals have equal contributions to the variation in trace (**A**_11_*), which was remarkable especially when the pedigree depth was shallow and limited information was available to construct the **A** matrix. There was little difference in the number of cows selected in each iteration among MCA methods using A3, A4, and A5. This was due to there being little difference between matrices A3, A4, and A5, as shown in Table 2. Many of the 575 cows were the ones introduced from the markets. Therefore, in this study, a cow whose descendants were among the 575 cows seems to have had a higher possibility of being selected as a candidate by MCA.

Meanwhile, all MCG methods selected one cow per iteration. Unlike **A**, **G** contains information on Mendelian sampling. **G** also seems to capture older relationships compared to **A** [41]. In this study, both GV1 and GV2 contained the most detailed information and therefore MCG could always select one cow per iteration.

### 3.3. Differences in Selected Cows

For MCA, the results were more similar to those with the A5 matrix when the numbers of generations traced back were larger (Figure 2). This was expected, because the block for the 575 cows of the **A** matrix for MCA became more similar when the number of generations traced back was similar (Table 1). The selected cows were not completely the same between the MCG methods with GV1 and GV2 (Figure 3). The number of commonly selected cows between the MCA and MCG methods was generally lower than that between different MCA methods or between different MCG methods (Figure 2 and Figure 3). These results imply that, at least in this study, differences in the information (pedigree or SNP markers) used to calculate the relationship matrix and the method used to calculate the matrix (the number of generations traced back for the **A** matrix and VanRaden’s methods 1 and 2 for the **G** matrix) affected the results.

### 3.4. General Discussion

Using high-density SNP genotypes for many individuals, generally several thousand or more, is expected to improve accuracy of GP and increase the number of SNP markers detected in GWAS that are significantly associated with traits. However, constructing a large-sized population, in which all individuals have real high-density genotypes seems to be too costly. Therefore, genotype imputation from low-density genotypes to high-density genotypes is a possible counterpart for cost-effective GP and GWAS. Genotype imputation has become available for Japanese Black cattle populations at commercial-chip level, such as imputation from the Illumina LD to 50 K v2 genotypes or from the 50 K v2 to HD genotypes (e.g., [21,23,24]). In the future, whole-genome sequence data will become available for Japanese Black cattle. This could bring the opportunity to perform GP and GWAS with much denser DNA markers in this breed and more ability to re-genotype some individuals to construct a reference population for genotype imputation from commercial SNP genotypes to sequence-level genotypes. Therefore, it is important to study methods of selecting candidates to be re-genotyped in order to construct a reference population for genotype imputation in Japanese Black cattle.

There is no study applying MCA and MCG methods to Japanese Black cattle populations. Furthermore, as far as we know, no previous study has assessed the effects of the content of pedigree information used to calculate the **A** matrix on the MCA method and the **G** matrix on the MCG method. Here, we assessed the effect of the pedigree depth on the MCA method, as well as the effect of the method to calculate the **G** matrix (VanRaden’s methods 1 and 2, both of which have been used in previous studies on genomic prediction in Japanese Black cattle) on the MCG method using our Japanese Black donor cow population. We used the iterative procedure of Yu et al. [26] for candidate selection in the MCA and MCG methods. Recently, Pocrnic et al. [42] introduced this procedure for selecting core animals for the algorithm for proven and young (APY) [43]. The pedigree depth affected the number of cows selected as candidates per iteration by the MCA method (Figure 1). The 575 cows with SNP genotypes included the ones introduced from the markets and their (grand)daughters. The cows introduced from the markets might have information on up to three generations of ancestors, namely, sire, maternal grandsire, and great-grand maternal sire of a cow. Therefore, we set the maximum number of generations traced back as five. The number of common ancestors in the pedigree data decreased by decreasing the number of generations traced back, which could be the main reason why the results were different. It should be noted that the pedigree depth for our population could be shallower than the pedigree data used in previous studies for Japanese Black cattle (e.g., [4,34,35,44]). Furthermore, the pedigree structure of a livestock population appears to be affected by the content of the selection scheme and mating plan. Thus, the performance of the MCA method is different in each population.

The MCG method with the **G** matrix calculated using commercial SNP markers appears to be more reasonable than MCA to construct a refence population for genotype imputation. We collected genotype information on genome-wide SNP markers for our Japanese Black donor cows using Illumina BovineSNP50 BeadChip in order to perform GP and GWAS for superovulatory response traits. Using higher-density SNP chips might give improved performance of GP and GWAS due to increased SNP markers (which involve (much) higher cost) to obtain genotype information of all cows. To overcome this, we are now planning to re-genotype some cows as a reference population of genotype imputation to higher-density SNP genotypes. We believe that the findings in this study could contribute to the future high performance of GP and GWAS with accurately imputed higher-density SNP genotype information. Meanwhile, the selected cows did not completely match between GV1 and GV2 when the same number of cows was selected (Figure 3). McEwin et al. [27] used the **G** matrix calculated by VanRaden’s method 1 [28], while Yu et al. [26] used the **G** matrix calculated using the method of Meuwissen et al. [45]. Many other methods of computing the relational matrix using genome-wide markers have been proposed (e.g., [28,46,47]), and further study might be valuable to select a suitable method to calculate the relationship matrix for the MCG method.

Previous studies for Japanese Black cattle populations have used the Beagle software, which implement population-based algorithm without pedigree data, to perform genotype imputation (e.g., [21,24,29]). We have also imputed missing genotypes for the 575 cows using the Beagle, and this study assumed the use of the Beagle for genotype imputation to high-density genotype information. Under this situation, the MCG method might be a reasonable choice for candidate selection. On the other hand, there is also software, including LDMIP [48] and FImpute [49], also utilizing family-based algorithm with pedigree data, which can impute genotypes. One possible merit of higher-density genotype information is the use of rare variants. It has been reported that imputation accuracy for rare variants was greater with software utilizing family-based algorithm than software with population-based algorithm [49,50] and that pedigree-based relatedness between target individuals and reference individuals had more impact on imputation accuracy with family-based algorithm than with population-based algorithm [51,52]. The SNP ascertainment bias might also affect the results [53]. These findings gave us the motivation to use the MCA method in this study. The **A** matrix is expected to contain more detailed information on additive genetic relatedness among the individuals when the pedigree is deeper and more complete. On the other hand, very deep pedigree might not be necessary especially when the genetic distance between target individuals and reference individuals is small—for instance reference individuals are the (grand)parents of target individuals. This supports the motivation to use matrices differing in the number of generations traced back, in this study. Making the information on rare variants available is expected not only to improve the performance of GP and GWAS but also to contribute better management of genetic diversity of a population [54]. A more sophisticated study for using high-density genotype information, such as sequence data, in GP, GWAS, and managing the genetic diversity of Japanese Black cattle is warranted.

## 4. Conclusions

We compared the similarity between the results of the MCA and MCG methods proposed by Yu et al. [26] to select individuals as candidates of a reference population for genotype imputation in in a Japanese Black cow population. Different **A** (pedigree depth) and **G** (method of calculation) were used for MCA and MCG, respectively. MCG selected one cow per iteration, while MCA sometimes selected multiple cows, especially when the number of generations traced back was small. Our results suggest that the results of the MCA and MCG methods depend on the content of the pedigree information used to calculate the **A** matrix and the method to calculate **G** matrix.

## Figures and Tables

**Figure 1 animals-13-00638-f001:**
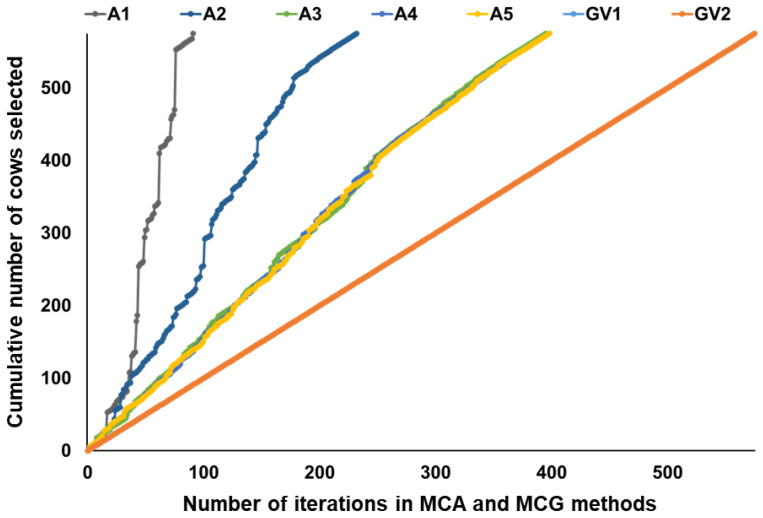
Changes in cumulative number of cows selected as candidates by MCA and MCG methods.

**Figure 2 animals-13-00638-f002:**
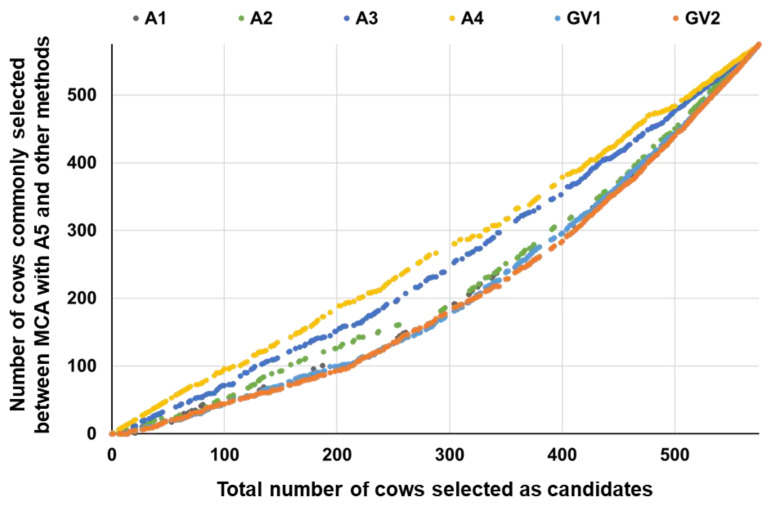
Changes in number of cows commonly selected between MCA with A5 matrix and other methods.

**Figure 3 animals-13-00638-f003:**
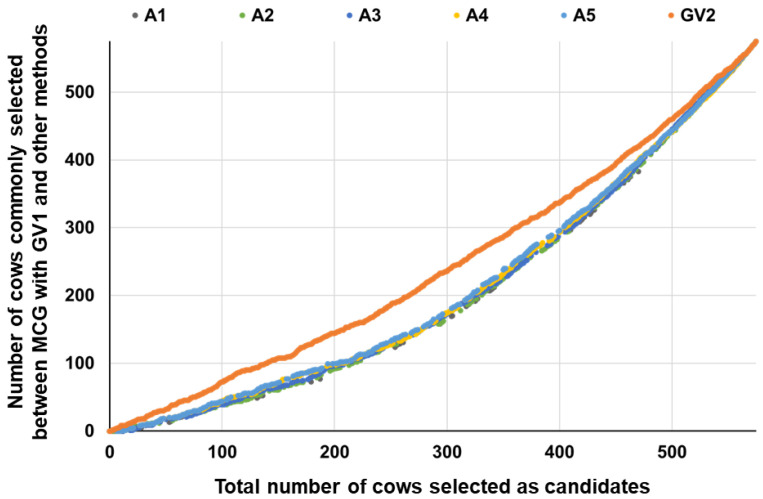
Changes in number of cows commonly selected between MCG with GV1 matrix and other methods.

**Table 1 animals-13-00638-t001:** Number of animals in each generation traced back from the 1874 cows.

Number of Generations Traced Back	0	1	2	3	4	5	Total
Number of animals	1874	1521	1129	141	17	2	4684

**Table 2 animals-13-00638-t002:** Basic statistics of the elements of the seven relationship matrices for the 575 cows.

Matrix	Diagonal Elements				Upper-Triangular Elements			
	Mean	SD	Min	Max	Mean	SD	Min	Max
	Pedigree-based additive genetic relationship matrix (**A** matrix)							
A1	1	0	1	1	0.025	0.075	0	0.5
A2	1.005	0.024	1	1.125	0.067	0.087	0	0.625
A3	1.012	0.030	1	1.156	0.084	0.090	0	0.625
A4	1.014	0.032	1	1.172	0.088	0.092	0	0.641
A5	1.014	0.032	1	1.172	0.088	0.092	0	0.641
	Genomic relationship matrix (**G** matrix)							
GV1	0.967	0.080	0.785	1.189	0.003	0.087	−0.227	0.613
GV2	0.970	0.130	0.725	1.409	0.003	0.080	−0.189	0.651

SD: standard deviation; Min: minimum value; Max: maximum value.

## Data Availability

The data supporting the findings of this study are shown in the manuscript.

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
