# Peer review of "Comparing Methods to Select Candidates for Re-Genotyping to Impute Higher-Density Genotype Data in a Japanese Black Cattle Population: A Case Study"

_animals, 2023, doi:10.3390/ani13040638_

Round 1

Reviewer 1 Report

Good read, interesting, very good writing. I missed a bit on the  justification and cost/benefit analisys of the proposed scheme, to be included in the very brief introduction.

There is a couple of missing "the" articles when refering to matrices at the very end of the paper.

Reviewer 2 Report

It is an interesting study on an important topic. However, one thing that should be clarified is that although the number of cows commonly selected by different methods is measured, the performances of the methods are not comparable. This is simply because the performance of each method is measured (trace(A11*)) on a matrix different from the other methods. For example, there is no evidence of whether GV1 is better or GV2. This study shows the differences but does not provide evidence of which method is better.

Because MCG could select one cow per iteration does not mean it is better than MCA. MCA could select multiple cows in some iterations, which MCG could not. This also does not mean that MCA is better than MCG. Generally, for genomic selection, it is better to rely on genomic information rather than pedigree.

Another major comment is that the motivation behind GV1PCA50 and GV1PCA90 is not clear. Why someone give up on G and go for GV1PCA50 or GV1PCA90? There is no justification for it. Similarly, it is not mentioned that if genomic information is available, why someone may prefer using “pedigree information for genomic selection” rather than “genomic information for genomic selection”. I suggest discarding GV1PCA50 and GV1PCA90 from the manuscript as those do not have a strong justification.

Please see below for other major and minor comments.

Simple summary: please mention the main difference between MCA and MCG.

L13: This line is not informative.

L15-16: Change “and how the genomic relationship matrix (G matrix) was calculated” to “and the method of calculating the genomic relationship matrix (G matrix)”.

L16-18: this sentence repeats the information in the previous sentence.

L50: Change “difficult” to “difficult and expensive”.

L52: Here and elsewhere, change “lower-density” to “low-density”.

L52: Change “information” to “to high-density genotype”.

L55: Change “imputed” to “imputing”.

L56: “lower-density SNP genotypes” Please mention the density (50K?).

L58-59: “and that with genotype information, included imputed ones” This makes the sentence hard to read. Please re-write the sentence.

L70: Change “the performance of a strategy” to “strategies”.

L71: Change “(e.g., [25–27]), and Yu et al. [26]” to “(e.g., [25–27]). Yu et al. [26]”.

L72: “the conditional variance” Which variance (additive genetic variance)? Conditional to what? Please be more specific.

L72: This is too little information to introduce MCA and MCG. What do the abbreviations stand for?

L74: Here and elsewhere, matrices A and G should be presented as bold.

L75: Change “a population” to “the population”.

L94,96: What is the difference between 54609 and 52524?

L100: Change “Beagle 3.3.2 software” to “Beagle v3.3.2”.

L101: Change “The number of generations traced back was up to five from” to “Pedigreed was traced back up to five generations from the”.

L103: Please Provide the number of animals with both parents missing, the number of animals in F0, the number of animals with sire-only missing, and the number of animals with dam-only missing (in the results).

L105: Change “A” to “block of A”.

L117: I guess you mean “50% and 90% of the variation”. The sentence is incorrect. All the 575 genotypes explain more than (exceeding) 50% and 90% of the total variation.

L128-135: This section is redundant and unnecessary. Of course, you will get different diagonal values that are worth testing rather than equalizing to 1. Yes, you set all the diagonal values to 1, but you deliberately change the off-diagonal values. There is no need to test such a hypothesis, and there is no interest in using a correlation matrix (CR) rather than a covariance matrix (G).

L141: “A matrix” A is different for A1-5. Which A did you use for the above equation when it comes to A1-5? If different, then you cannot compare A1-5. For the same reason, MCA and MCG are not comparable. However, theoretically, MCG is better than MCA, because it relies on genomic information for genomic selection.

L147: Change “G matrix was used instead of A matrix” to “the G matrix was used instead of the A matrix”.

L158: Change “the setting of the cumulative contribution ratio” to “reducing the cumulative variance in G”.

169-172: No, this is because some animals have equal contributions to the variation in A.

L176-182: Please re-write these lines. What is written is grammatically incorrect and does not convey the message.

Figures 3,4: Please modify the y-axis text.

L195: “submatrix of A5” or “submatrix of A”?

L201: Here and elsewhere, change A11 to A11*.

Figure 5: Given the similarities, this figure is difficult to observe. If A11 is different for A1-5, it is not clear what this figure conveys, and what is the conclusion about it. Yes, for A1-5 at a given MCA iteration, a similar proportion of their trace(A11*) is reduced, but there is nothing interesting or novel about it.

L211: Change “trace” to “trace(A11*)”.

L213: Change “study” to “study (Figure 8)”.

L214: Where are the results (GV1 vs GV2)?

Figure 6: These lines are so similar that you may omit this figure by mentioning the similarity in a sentence.

General discussion: Please remind the reader what you mean by performance.

L275: “the degree of effect appeared to differ among the matrices used (Figure 8)”. This figure shows the number of common cows found by different methods. It does not show any effect. It is not clear which effect of what on what.

Reviewer 3 Report

In order to the manuscript (animals-2137894) with tittle “Performance of methods to select candidates for re-genotyping to impute higher-density genotype data in a Japanese Black cattle population: a case study”.

I would like to share some major comments about it:

I suggest including the pedigree completeness used to obtain the A Matrices. This issue must be taken into account due to results and discussion are influenced by pedigree completeness (4686 animals’ line 102-103). Please, also do it from A1 to A5 (Line 107-108).

Inconsistency in the number of individuals include in the pedigree 4686 (line 103) vs 4684 (Line 106).

Please describe relevant details of the algorithm or methods used by BLUPF90+ software to build A Matrices.

Authors must descried the equations to obtain G matrices. It is not clear in the manuscript the reason why the authors just include G-Matrix 1 and 2 from Van-Raden. Another’s G matrices are also available such as G-Matrix proposed by Yang.

In material and methods, there is no reason to support the idea to perform principal component analysis (PCA) just to GV1. Please explain the reason; why was performed PCA only to GV1? If authors consider it, please include PCA on GV2.

As far, I understand the cumulative contributions to the total variation of 50% and 90% are quite arbitrary. The value of 70 or 75% is a common criterion for PCA. If authors consider it, please include it.

Round 2

Reviewer 2 Report

Thanks to the authors for making a good revision. Lots of improvements have been made.

The word "performance" is usually associated with "low performance" vs "high performance". This study mainly considers similarities between methods. However, I cannot find any word better than performance! If the authors know any better word, they may use it instead of "performance".

I suggest changing the manuscript to a "Short Communication". It is a short study, and the similarities are compared, not the performances.

The motivation behind MCA is unclear. The authors need to mention why they considered using pedigree instead of genotype information for "re-genotyping".

The motivation behind A1-5 is also unclear. It seems to me that the authors compared some scenarios not because those were good alternatives, but for having multiple scenarios that they could compare with each other.

Due to the style of the tracked changes, I found it difficult to review the manuscript. Please send the next revision with the accepted changes.

Round 3

Reviewer 2 Report

IMPORTANT: Tables and figures should be independent of the body text. Any notation or abbreviation in them should be defined in the footnote.

The writing should be improved.

L103: Change "that were genotyped with the Illumina BovineSNP50 v2 BeadChip was used in this study" to "on the Illumina BovineSNP50 v2 BeadChip was used"

L125: Delete "for the 575 cows". Already mentioned in the previous line.

L136: Change "the block for the 575 cows of A matrix" to "the block A for the 575 cows"

L143: Change "1,521 animals newly appeared" to "1,521 parents (Table 1)"

L161: Delete "while whose parents were treated as missing". It is not necessary and it is confusing.

L184: Rephase "our population"

L187: Delete ", also as expected"

L191: Change "judged" to "concluded"

L256: Change "were ones" to "were the ones"

L257-258: Thes lines are difficult to read and understand.

L259: Delete "in this study"

L260: Change "do" to "does"

L259-262: Change "The G matrix generally contains information on Mendelian sampling while the A matrix do not contain, and the relationship captured by the G matrix seems to be older than that captured by the A matrix" to "Unlike A, G contains information on Mendelian sampling. Also, G seems to capture older relationships compared to A"

L262: Change "Therefore, both" to "Both"

L268: Change this line to "For MCA,the results were more similar to those with the"

L271: Change "Selected cows did" to "The selected cows were"

L272: Delete "in this study"

L273-278: It is very obvious that the differences between MCA and MCG are greater than those within MCA and MCG. Also, the conclusion on that is incorrect.

L293-294: "50K" or "Illumina 50K" to be more specific?

L305: Change "pedigree depth (the number of generations traced back)" to "pedigree depth"

L313: Change "included ones" to "included the ones"

L316-318: Change "When the number of generations traced back decreased, the number of common ancestors in the pedigree data for constructing the A matrix decreased" to "the number of common ancestors in the pedigree data decreased by decreasing the number of generations traced back"

L323-324: Change "should be assessed in each population" to "is different in each population"

L326: Change "among the MCG with different G matrices" to "between GV1 and GV2"

L327: "when the number of cows was the same". I cannot understand what you mean.

L356: Change "result" to "results"

L367: Change "deeper" to "deeper and more complete", and "much deep pedigree" to "very deep pedigree"

L369: Change "is" to "are"

370: Change "differing the" to "differing in the"

370-371: Here and elsewhere change "in this study." to ", in this study."

377: Change "We assessed the performance of" to "We compared the similarity between the results of"

L377-380: This is a nearly 4 lines sentence. Please change L379-380 to "in a Japanese Black cow population. Different A (pedigree depth) and G (method of calculation) were used for MCA and MCG, respectively. MCG selected one cow per iteration, while"
